# An Update on MYBPC3 Gene Mutation in Hypertrophic Cardiomyopathy

**DOI:** 10.3390/ijms241310510

**Published:** 2023-06-22

**Authors:** Bogdan-Sorin Tudurachi, Alexandra Zăvoi, Andreea Leonte, Laura Țăpoi, Carina Ureche, Silviu Gabriel Bîrgoan, Traian Chiuariu, Larisa Anghel, Rodica Radu, Radu Andy Sascău, Cristian Stătescu

**Affiliations:** 1Department of Internal Medicine, Faculty of Medicine, Grigore T. Popa University of Medicine and Pharmacy of Iasi, 16 University Street, 700115 Iasi, Romania; bogdan-sorin.tudurachi@d.umfiasi.ro (B.-S.T.); laura.tapoi@gmail.com (L.Ț.); carina.ureche@yahoo.com (C.U.); larisa.anghel@umfiasi.ro (L.A.); rodica.radu@umfiasi.ro (R.R.); radu.sascau@umfiasi.ro (R.A.S.); cristian.statescu@umfiasi.ro (C.S.); 2Prof. Dr. George I.M. Georgescu Institute of Cardiovascular Diseases, Carol I Boulevard, No. 50, 700503 Iasi, Romania; leonteandreea32@gmail.com (A.L.); birgoan_gabi@yahoo.com (S.G.B.); traian.chiuariu@gmail.com (T.C.)

**Keywords:** hypertrophic cardiomyopathy (HCM), MYBPC3, MYH7, genetic mutations

## Abstract

Hypertrophic cardiomyopathy (HCM) is the most prevalent genetically inherited cardiomyopathy that follows an autosomal dominant inheritance pattern. The majority of HCM cases can be attributed to mutation of the MYBPC3 gene, which encodes cMyBP-C, a crucial structural protein of the cardiac muscle. The manifestation of HCM’s morphological, histological, and clinical symptoms is subject to the complex interplay of various determinants, including genetic mutation and environmental factors. Approximately half of MYBPC3 mutations give rise to truncated protein products, while the remaining mutations cause insertion/deletion, frameshift, or missense mutations of single amino acids. In addition, the onset of HCM may be attributed to disturbances in the protein and transcript quality control systems, namely, the ubiquitin–proteasome system and nonsense-mediated RNA dysfunctions. The aforementioned genetic modifications, which appear to be associated with unfavorable lifelong outcomes and are largely influenced by the type of mutation, exhibit a unique array of clinical manifestations ranging from asymptomatic to arrhythmic syncope and even sudden cardiac death. Although the current understanding of the MYBPC3 mutation does not comprehensively explain the varied phenotypic manifestations witnessed in patients with HCM, patients with pathogenic MYBPC3 mutations can exhibit an array of clinical manifestations ranging from asymptomatic to advanced heart failure and sudden cardiac death, leading to a higher rate of adverse clinical outcomes. This review focuses on MYBPC3 mutation and its characteristics as a prognostic determinant for disease onset and related clinical consequences in HCM.

## 1. Introduction

Hypertrophic cardiomyopathy (HCM) represents the most prevalent cardiomyopathy worldwide; its prevalence varies greatly, ranging from 0.02 to 0.2% in western and Asian nations [1], with an estimated frequency in the general population between 1:200 and 1:500 [2]. It is often characterized by fluctuating degrees of diastolic dysfunction, thickening of the left ventricular wall, progressive heart failure, and sudden cardiac death [3]. A major HCM consequence that affects 20 to 30% of individuals is atrial fibrillation (AF), occurring far more frequently than in the general population. Patients with HCM have a 2.5-fold greater risk of mortality, a 2-fold increased risk of sudden death, a 2.5-fold increased risk of heart failure, a 3-fold increased risk of heart failure, and a 7-fold increased risk of thrombosis when they have AF compared to those in sinus rhythm [4].

HCM is transmitted through an autosomal dominant inheritance pattern. Two sarcomere proteins—myosin-binding protein C (MYBPC3) and myosin heavy chain (MYH7)—are involved in 70% of instances of sarcomere gene mutation-positive HCM, even though mutations of several proteins can result in it. MYBPC3 is the primary gene known to cause restrictive cardiomyopathy, dilated cardiomyopathy, and left ventricular non-compaction [5,6,7]. The human MYBPC3 gene’s structure and sequence were established by Carrier et al. in 1997 [8]. The primary structural protein of the heart muscle, cMyBP-C, is encoded by the MYBPC3 gene. It interacts with actin, myosin, and titin to maintain the integrity of the sarcomeric layer [9,10].

The complex interplay of various determinants, including the underlying genetic mutation and environmental factors, contributes to the morphological, histological, and clinical manifestations of HCM. While the causal mutation is a key factor and necessary for the phenotype to develop, the histological and clinical phenotype may be influenced by genetic background, including by the presence of other pathogenic variants involved in cardiac hypertrophy pathways as well as epigenetic and environmental factors.

This review focuses on the role of the MYBPC3 gene in the pathophysiology of HCM in correlation with the clinical phenotype of this mutation as an update for this specific gene.

## 2. Structure of the MYBPC3 Gene

Adult human muscle contains three very similar MyBP-C isoforms: the slow skeletal isoform, which is expressed by the MYBPC1 gene on chromosome 12q23.3; the fast skeletal isoform, which is produced by the MYBPC2 gene on chromosome 19q33.3; and the cardiac isoform, cMyBP-C [11]. With its eight immunoglobulin-like and three fibronectin-like domains and distinctive structural differences from the slow and fast skeletal isoforms, cMyBP-C is the perfect platform for signaling. These differences include an extra immunoglobulin-like domain (C0 do-main) at the N-terminus, many phosphorylation sites in the MyBP-C motif (M motif) between the immunoglobulin-like domains C1 and C2, and a 28-amino-acid insertion inside the C5 domain [12,13,14].

cMyBP-C represents a flexible rod-like protein situated in an anti-parallel orientation within the A-band of the cardiac sarcomere, and exhibits a robust binding affinity to the myosin filament backbone via its C-terminal domain. Additionally, its N-terminal domains bind to either the myosin head or the actin filament, thereby playing a crucial role in regulating the interaction between actin and myosin, the kinetics of muscle contraction and relaxation, and the sensitivity of myofilaments to calcium ions [15,16]. The schematic representation of cMyBP-C is illustrated in Figure 1.

## 3. MYBPC3 Mutations in HCM Patients

The impact of MYBPC3 mutation in HCM patients is important, as it is one of the most frequent mutations responsible for this disease. Recently, a total of 242 consecutive HCM index patients (127 males) were examined using next-generation sequencing utilizing a specially constructed gene-panel that included 98 genes relevant to cardiomyopathy, demonstrating a detection rate comparable to prior data. Ninety patients in total (37%) carried P/LP variants which are pathogenic or likely to be pathogenic, and 93% of patients had P/LP variations in genes that were positively associated with HCM. Most of these individuals had mutations in MYH7 (23%) and MYBPC3 (61%), with the majority of these mutations being frameshift, nonsense, or splice variants [17].

### 3.1. Truncating Mutations

Although cMyBPC mutations are a major contributor to HCM, the exact molecular basis of these mutations is as yet unknown. Various molecular mechanisms have been suggested as the pathogenesis underlying hypertrophic cardiomyopathy (HCM) caused by truncated variants of MYBPC3. Reduced cMyBP-C levels (protein haploinsufficiency) can occur either due to the mutant not being created or because of it breaking down rapidly, a state in which the gene product of the normal allele is incapable of compensating for the reduced product originating from the mutant allele. Additionally, if the mutant protein with the shortened amino acid is produced and becomes dominant, it may operate as a “poison peptide” by exerting a dominant-negative (DN) influence and modifying the structure and function of the sarcomere [18,19,20].

According to currently available data, around 50% of MYBPC3 mutations lead to shortened protein products, whereas the other mutations result in insertion/deletion, frameshift, or single-amino-acid missense changes [21,22,23]. Ninety percent of the mutations in MYBPC3 that cause premature termination codons (PTCs) on one allele in HCM patients are heterozygous frameshift, nonsense, or splice site alterations that lead to truncated polypeptides [24]. Therefore, it is believed that these mutations cause HCM by causing an allelic loss of function via nonsense-mediated RNA decay (NMD) of transcripts that include PTC, indicating that the majority of the mutant transcript is degraded, which causes a decrease in MyBP-C concentration in the sarcomere [15,24]. However, it has been demonstrated in the past that mRNA from the mutant allele of MYBPC3 is detected in sufficient amounts in the myocardium of individuals with MYBPC3-linked HCM to serve as a model for mutant protein production. The absence of truncated proteins indicates that the condition is caused by haploinsufficiency of full-length MYBPC3 within the sarcomere, although the production of mutant proteins has not been completely ruled out as a possible cause. Expression of truncated mutant MYBPC3 protein has been linked to decreased functionality of the ubiquitin–proteasome system (UPS), one of the main cellular protein degradation processes, in multiple investigations employing both cell and animal models [25]. Truncating mutations commonly trigger the activation of the NMD and UPS, thereby leading to haploinsufficiency of the MYBPC3 protein [26].

The myocardium of patients with HCM who possess heterozygous truncating mutations in MYBPC3 did not exhibit any truncated proteins. Additionally, the quantity of wild-type (WT) cMyBP-C present in these patients was approximately 30% lower than that observed in the myocardium of non-failing donors, suggesting that haploinsufficiency is the primary mechanism underlying the development of disease in the context of truncating mutations. Conversely, missense mutations yield alterations in amino acids, leading to the formation of stable full-length mutant proteins that are nearly indistinguishable from the wild-type protein [16,18,27,28].

PTCs, which are caused by the majority of MYBPC3 mutations, cause RNA degradation and a decrease in MyBP-C in the hearts of HCM patients. However, induced pluripotent stem cell cardiomyocytes (iPSCMs) with MYBPC3 mutation have not consistently shown a decrease in MyBP-C. Helms et al. [15] employed patient-derived and genome-engineered iPSCMs to assess the early impact of the MYBPC3 mutation. Comparing iPSCMs with frameshift mutations to those with MYBPC3 promoter and translational start site deletions allowed researchers to conclude that the main contributing factor to PTCs is allelic loss of function. There was a decrease in wild-type mRNA in all heterozygous iPSCMs, however, there was no decrease in MyBP-C protein, showing protein-level compensation via what was then an unidentified mechanism. In the presence of compensatory MyBP-C levels, heterozygous mutant iPSCMs had normal contractile properties despite homozygous mutant iPSCMs displaying contractile dysregulation. The process of MyBP-C degradation was slower in heterozygous mutant iPSCMs, although the rate of MyBP-C synthesis remained lowered. These results demonstrate that despite MYBPC3 allelic loss of function driven by truncating mutations, cardiomyocytes have an intrinsic ability to achieve normal MyBP-C stoichiometry [15]. In a comparable manner, another study found that MYBPC3 PTC mutation-carrying iPSC-CMs showed abnormal calcium signaling and molecular dysregulation even in the absence of significant MYBPC3 protein haploinsufficiency, directly linking the sustained activation of the NMD pathway to the occurrence of HCM disease and MYBPC3 PTC mutations associated with it [3].

The production of the functional protein should only be below the level required for appropriate function; it is not necessary for it to be 50% of the normal overall expression. These include processes of NMD and mutation-induced protein instability, which can responsible for allelic imbalance. In-frame mutations that cause loss of function, for example, might result in mutant proteins being present but inactive. The relative expression of wild-type vs. mutant proteins can be altered by allelic imbalance, potentially either aggravating or correcting for haploinsufficiency [19,29]. Therefore, the production of one active copy of MYBPC3 leads to cMyBP-C protein content that is inadequate and below the minimum level needed for it to perform normally within the sarcomere. Insufficient cMyBP-C may result in an imbalance in the stoichiometry of sarcomere proteins, which may in turn cause dysgenesis and decreased contractility, as cMyBP-C co-expresses with other thick filament components in the early stages of myofibril formation [8,19,30]. Contractile dysfunction and HCM may be produced in vivo by overexpressing the cMyBP-C protein with a mutated C10 domain (cMyBP-C^ΔC10mut^). This protein does not integrate into the sarcomere’s C-zone, instead being localized to the cytosol and Z-line. These data grant credibility to the poison polypeptide theory, which holds that a rise in harmful mutant and truncated proteins causes a series of events that starts with inappropriate incorporation, this being considered another pathogenic mechanism in HCM. Contractile dysfunction is followed by cMyBP-C^ΔC10mut^, then pro-hypertrophic signaling and remodeling of the heart [31]. The results of both in vitro and in vivo investigations suggest that cardiomyocytes may express cMyBP-C-C10mut, which would have “poison polypeptide” effects on cardiac contractility [31,32]. According to research by De Lange et al. [33], although three-dimensional synthetic cardiac tissue made from heterozygous MYBPC3 null cardiomyocyte cell models expresses normal levels of cMyBP-C, the tissue exhibits haploinsufficiency. The same authors have presented results demonstrating abnormalities in calcium transient duration and tension development in two-dimensional culture with normal cMyBP-C levels, demonstrating that increased protein demand conceals haploinsufficiency [33]. The current consensus is that MYBPC3-truncating mutations lead to either no translation or to quick destruction of the truncated protein, excluding the harmful truncated protein [19,34,35]. Another potential harmful effect that has been identified by research as a sign of cMyBP-C haploinsufficiency is myosin levels that are less super-relaxed. When myosin is in its super-relaxed form, it requires cMyBP-C and consumes less power. Super-relaxed myosin levels were found to be decreased in HCM patients with MYBPC3 mutation causing haploinsufficiency [36,37]. In MyBPC haploinsufficiency, myosin dysfunction is a key pathophysiologic mechanism of HCM. Myosin conformations become unstable as a result of cMyBPC truncation, resulting in harmful ratios of disordered relaxation/super-relaxation (DRX/SRX) conformations that promote hypercontractility, delay relaxation, and increase energy usage [38].

The NMD mechanism, which degrades and drastically reduces the quantity of nonsense mRNAs, mostly regulates the expression of truncating MYBPC3 mutations at the mRNA level. Mutant proteins are removed through selective destruction of mutant proteins by the UPS. When the truncating mutation is mono-allelic, the production of the wild-type allele partially compensates for the absence of mutant proteins, causing haploinsufficiency of the MYBPC3 protein (50–70% of wild-type levels) and the distinctive symptoms of HCM. The degradation of mutant proteins and nonsense mRNAs of MYBPC3 caused by bi-allelic truncating mutations reduces MYBPC3 function in the sarcomere [2].

RNA splicing is a post-transcriptional process that connects protein-coding exon regions and eliminates non-coding intron sequences from original RNA transcripts to create messenger RNA (mRNA). Frameshifts in MYBPC3 are frequently brought on by abnormalities in RNA splicing. While variations in the well-conserved surrounding areas may potentially interfere with splicing, variations at the crucial splice-site dinucleotides almost always result in mistakes in RNA splicing. This has to be confirmed by study of the mRNA sequence, as the genetic etiology of HCM cannot yet be accurately predicted using in silico bioinformatic techniques [39,40,41].

Several strategies have been proposed for gene therapy, such as gene replacement therapy, gene inactivation therapy, and gene augmentation therapy. Gene replacement therapy involves the substitution of a nonfunctional gene with a functional and healthy gene, while gene inactivation therapy aims to disable a disease-causing gene [42].

The overexpression of complementary DNA (cDNA) as a means of gene replacement presents a highly appealing therapeutic approach for sarcomeric cardiomyopathies. Pre-clinical findings indicate that replacement of the deficient protein in cardiomyocytes for severe forms of HCM can be achieved through MYBPC3 gene replacement therapy using adeno-associated virus (AAV) vector transfer. This has been demonstrated through the use of HCM mouse models, both mouse and human engineered heart tissues, and human induced pluripotent stem cell-derived cardiomyocytes (hiPSC-CMs). Studies have revealed that the administration of full-length WT Mybpc3 cDNA via the AAV vector in homozygous Mybpc3-targeted knock-in mice can effectively hinder the onset of cardiac hypertrophy and dysfunction by augmenting cMyBP-C protein levels. Additionally, expression of the native mutant alleles was observed to be suppressed [43,44].

Similar to these results, adenoviral-mediated MYBPC3 gene delivery over 7 days brought back cMyBP-C protein levels to WT levels and prevented hypertrophy in car-diomyocytes made from human embryonic stem cells (hESCs) [45]. It is anticipated that the introduction of a sarcomeric protein through exogenous means will serve to partially or fully supplant the native counterpart [46]. This is due to the fact that the sarcomere operates as a closely regulated system, with all structural components maintaining a consistent stoichiometry. The present investigation pertains to a study on the control cell line, wherein transfer of the MYBPC3 gene led to a 2.4-fold increase in the level of MYBPC3 mRNA while the level of full-length cMyBP-C protein remained unaltered. Transfer of the MYBPC3 gene elicited a 2.6-fold increase in the quantity of MYBPC3 messenger RNA in hypertrophic cardiomyopathy cells. Significantly, the cMyBP-C quantity in HCM CMs following gene transfer attained 81% of the quantity in the control cell line owing to the fact that the basal level was lower in the former. It has been hypothesized that the disease phenotype could be partially corrected under these circumstances, based on previous findings demonstrating partial restoration of the disease phenotype in engineered heart tissue derived from Mybpc3-deficient mice through MYBPC3 gene transfer, resulting in 80% restoration of cMyBP-C levels [16,47]. These findings suggest that the suppression of hypertrophy in HCM CMs can be achieved through partial restoration of cMyBP-C haploinsufficiency.

The incorporation of exogenous FLAG-cMyBP-C proteins into the sarcomere was observed to be proper in both cell lines. The replacement of the MYBPC3 gene exhibited a favorable impact on the mRNA levels of proteins linked to hypertrophy and calcium handling. For instance, after only seven days of MYBPC3 treatment, serum response factor (SRF) and S100A4, both of which are known to be elevated in cardiomyopathies, drastically decreased [48,49].

### 3.2. Nontruncating Mutations

In contrast to truncating MYBPC3 variants, nontruncating pathogenic variants such as missense and small in-frame deletion/insertion variations) make up around 15% of MYBPC3 HCM. Uncertainty surrounds the mechanisms behind MYBPC3 nontruncating pathogenic variations, as well as around whether individuals with missense mutations exhibit distinct phenotypic manifestations and clinical outcomes. In a clinical evaluation of MYBPC3 missense mutations the total production of proteins decreased, indicating that certain missense variations may enhance susceptibility to degradation by nonsense-mediated mRNA decay, ubiquitin-mediated proteolysis, and other mechanisms. Recent research has demonstrated that MYBPC3 missense mutations result in full-length mutant proteins, some of which appear to cause haploinsufficiency, while others do not [28,30,35,50,51]. Missense single nucleotide variants account for a significant proportion of presently identified human genetic disorders by causing the replacement of a solitary amino acid residue at the protein level. Certain MYBPC3 missense mutations and truncations can lead to haploinsufficiency [52,53]. Using functional investigations, it may be possible to better judge the pathogenicity of nontruncating MYBPC3 mutations and increase the number of findings from gene tests that may be used in clinical settings [54,55]. According to the level of structural stability, missense mutations are assumed to have a variety of pathogenic mechanisms, each of which is unique to the protein that has undergone the mutation. According to a study that analyzed the mechanism by which the pathogenic MYBPC3 missense mutation Y235S results in contractile dysfunction, the introduction of the Y235S missense mutation via lentiviral-mediated gene transfer did not result in any appreciable changes in the levels of protein expression or phosphorylation of cMyBPC or other sarcomeric proteins. Therefore, utilizing molecular dynamics stimulation (MDS), it was demonstrated that the expression of Y235S cMyBPC interferes with the C1 domain of cMyBPC’s ability to bind with myosin and actin. Y235S cMyBPC expression interferes with the inter- and intramolecular interactions of cMyBPC’s C1 domain with myosin and actin using MDS, speeding up the recruitment of new crossbridges (XB) to empty myosin binding sites on actin and transition XBs to force-bearing states. These findings demonstrate that a single missense mutation in the cMyBPC C1 domain changes the rate of XB cycling and the overall contractile behavior of cardiac muscle by destabilizing crucial residues involved in actomyosin interaction, and affects actomyosin binding by altering intramolecular interactions, domain stability/structure, and surface electrostatic potentials. Major increases in ATP turnover rates at the myofibrillar level lead to accelerated XB recruitment and dissociation rates, which lower the XB duty ratio. Chronically high ATP turnover rates at the whole-heart level cause myocardial energy deficits that partially explain HCM-induced in vivo contractile failure, hypertrophy, and diastolic dysfunction [56].

While the pathomechanisms involving non-truncating MYBPC3 mutations remain unknown, truncating variants have typically been documented as causing illness through protein haploinsufficiency. In this same regard, it is possible that nontruncating variants could worsen illness by reducing overall cMyBP-C content.

The two main mechanisms of protein haploinsufficiency caused by nontruncating variations have been described as changes in RNA processing and protein destabilization. Surprisingly, nontruncating pathogenic mutations may be harmful due to cMyBP-C haploinsufficiency as well, considering that their clinical presentations are identical to those of truncating variants. Because both truncating and nontruncating pathogenic variants have comparable clinical symptoms, it is probable that cMyBP-C haploinsufficiency contributes to the pathogenicity of certain nontruncating mutations as well [20,54,57].

### 3.3. Ubiquitin–Proteasome System and Nonsense-Mediated RNA Dysfunctions

All the eukaryotes that have been studied to date share the evolutionary conservation of the NMD surveillance mechanism. NMD quickly degrades PTC-containing transcripts, shielding the organism from the harmful consequences of C-terminally shortened proteins that occur as a result of dominant-negative or gain-of-function mutations [58]. NMD is not always effective, and transcripts that have a PTC extremely close to their 30′ terminus may avoid being specifically degraded. Avoiding this and other transcript quality control systems, such as the No-Go mRNA decay pathway, which ceases translation in ribosomes and encourages RNA decay through endonucleolytic pathways, can explain the common finding of low levels of nonsense mutant transcripts in human HCM myectomy samples [20,59,60,61]. Quality monitoring for abnormal proteins exists as well, and is similar to NMD for nonsense mRNAs. A large number of cytosolic, nuclear, and myofibrillar proteins are degraded by the UPS, whereas membrane proteins and organelles are broken down by lysosomes through autophagy. The UPS plays a key role in preventing the buildup of damaged, misfolded, and mutant proteins, and additionally controls transcription and regulates cell death [62]. Both in vitro and in vivo mouse transgenic models have revealed that the UPS is involved in the clearance of aberrant truncated cMyBP-C. The UPS function may be hampered by the buildup of mutant proteins with misfolded structures. In this context, it has been suggested that protein aggregates may overload the proteasome, obstructing the enzyme’s ability to perform its proteolytic function. Additionally, as people age, the effectiveness of their UPS system declines, which might account for the frequently late onset of HCM in MYBPC3 carriers. Integrating these two outcomes, a study comparing UPS activity in two distinct mouse models—a homozygous Mybpc3-targeted knock-in (KI) model expressing 10% truncated protein and a homozygous Mybpc3-targeted KO mouse expressing no cMyBP-C—found that after one year of life only the mice expressing the truncated protein displayed specific UPS impairment. Additionally, proteasome dysfunction was seen in the homozygous KI mice and correlated with the degree of left ventricular hypertrophy (LVH), showing that UPS dysfunction plays a role in the pathogenesis of HCM [30,63,64,65,66].

### 3.4. Allelic Imbalance

A higher expression of mRNA or protein from one allele compared to the other can be referred to as an allelic imbalance that differs from the normal 1:1 expression ratio. It has been established by allele-specific sequential investigation of gene expression tags that 25% of human genes exhibit allelic imbalance. Allelic imbalance can be generated through mechanisms beginning at the DNA, mRNA, and protein levels. The availability of a gene to transcription machinery can be affected by epigenetic alterations of one allele, such as gene imprinting or differential methylation/acetylation related to cis-acting abnormalities. By altering the 3’-untranslated region or microRNA binding, changes at the mRNA level might result in lower transcript stability. Furthermore, premature termination codon-containing mutant mRNA transcripts are vulnerable to NMD. If the mutation affects interaction with the protein quality control machinery, interactions between proteins, intrinsic folding of proteins, or cellular localization protein allelic imbalances can occur, often resulting in a reduction in stability of proteins due to mutation [19,67,68]. On the other side, a major mechanism leading to protein haploinsufficiency is the destruction of mutant transcripts and shortened proteins [20].

Allelic imbalance has been observed at the protein level in hypertrophic cardiomyopathy for only a few missense mutations in MYBPC3. Numerous MYBPC3 mutations that are pathogenic simply result in a single amino acid shift, providing mutant full-length cMyBP-C proteins that can integrate into sarcomeres at the same levels as the wild-type protein. This discrepancy may result from changes in the protein’s stability, the avidity of binding inside the sarcomere lattice, or asymmetry in allelic transcripts. Depending on the particular mutation, different samples have shown higher or lower percentages of mutant protein compared to the wild-type allele. These missense mutations’ molecular deficiencies are mainly unstudied. Many HCM-causing missense mutations, however, appear to function via alternate unexplained pathways. Several of these are assumed to impair cMyBP-C interaction with actomyosin filaments or to cause significant protein instability. It is unlikely that specific pathogenic MYBPC3 missense mutations induce HCM through conventional protein haploinsufficiency, as they maintain RNA splicing and protein thermal stability. These mutations are likely integrated into the sarcomere but are unable to operate effectively [24,35,50,69]. In research employing single-molecule force spectroscopy using atomic force microscopy (AFM) [70], it has been postulated that mutations that cause HCM may disrupt the nanomechanics of cMyBP-C, resulting in changes to sarcomere activity. As a result, it has been discovered that HCM-causing mutations can modify the mechanical stability and folding rate of the targeted domains, suggesting that modulation of cMyBP-C nanomechanics might have a role in the pathogenesis of HCM. A schematic representation of potential mutations that can appear at the MYBPC3 level is illustrated in Figure 2.

## 4. From Genetics to Clinical Implications

The presence of a large and well-characterized founder population of HCM patients carrying the MYBPC3:c772G>A variant offers a unique opportunity to study the pathogenesis of HCM in multiple human samples where the disease is caused by the same mutation. The c.772G>A variant was studied by Pioner et al. [71] for the first time by comparing the functional properties of ventricular cardiomyocytes and trabeculae from human samples of surgical origin (myectomy operation) with myocytes and engineered muscles differentiated from human induced pluripotent stem cells (hiPSC) obtained from the same patients. The findings of this study are consistent with the hypothesis that while compensatory slowing of calcium handling preserves contraction parameters, faster sarcomere kinetics decreases myocardial contraction. In fact, cardiomyocytes from myectomy samples and hiPSC exhibited secondary ion channel and calcium homeostasis alterations, suggesting an early adaptive response to primary sarcomeric changes brought on by haploinsufficiency. Considering that haploinsufficiency is the predominant pathogenic mechanism in almost all MYBPC3 mutations associated with HCM, recent studies indicate that this aspect may have implications for the broader cohort of individuals with MYBPC3 mutations, who constitute 20% to 30% of all HCM cases. Mavacamten, for example, normalizes cross-bridge kinetics and sarcomere energetics, which may prevent or lessen HCM-related pathology when provided to gene carriers in the early stages of the condition [71].

Each of these mutations has a unique clinical manifestation. For instance, afflicted males with the MYBPC3 c.2149-1G>A splicing variation have earlier onset of HCM and greater penetrance than women. When a mutation’s pathogenicity is confirmed, relatives who carry the mutation are eligible for monitoring to detect early signs of illness, treat it, and prevent its consequences [72].

Recent studies have suggested that mutation-positive HCM patients have certain characteristics that expose them to a higher risk of sudden death. They are often asymptomatic, usually diagnosed at a younger age, with a family history of HCM and sudden death; in addition, they more frequently exhibit arrhythmic syncope, severe asymmetrical left ventricular hypertrophy (≥20 mm), and absence of negative T waves in the lateral ECG leads. This could offer new insights into the development of a clinical score to predict genetic yields in patients with HCM [73,74].

### 4.1. Demography

There appear to be gender differences in patients with HCM and MYBPC3 mutations. In this sense, a study which included 61 subjects revealed that even though females experience disease onset at a later stage, they tend to display more severe symptoms at the time of diagnosis, and are more likely to experience heart failure events after developing hypertrophy [75].

A nationwide cohort of Icelandic individuals who carried the same MYBPC3 founder mutation underwent comprehensive family screening, which revealed that HCM is more likely to occur at a younger age in men. However, the lifetime risk of developing HCM appeared to be high and equal for both genders. Relatives who were diagnosed with HCM during the family screening process tended to be older at the time of their initial evaluation and to exhibit a less severe clinical phenotype compared to HCM probands. The phenotypic characteristics of the Icelandic MYBPC3 founder mutation were found to be strikingly similar to other populations with truncating MYBPC3 variants, suggesting a common effect resulting from haploinsufficiency in the myosin-binding protein [76].

Although historically linked to late-onset disease, MYBPC3 mutations can lead to HCM in the pediatric population as well. Similar to adults, the phenotypic expression in children is heterogenous, including both severe and arrhythmogenic phenotypes [77]. Furthermore, a substantial portion of HCM cases that occur during childhood are caused by compound genetic variants. The presence of bi-allelic truncating MYBPC3 mutations in patients is linked to neonatal cardiomyopathy, and can result in heart failure and mortality within the first year of life. These findings suggest that a gene-dosage effect could be responsible for the manifestation of the disease at younger ages [78,79,80].

### 4.2. Phenotype

HCM caused by MYBPC3 mutations can have variable clinical presentations ranging from mild to severe, and may be associated with an increased risk of sudden cardiac death, although not all individuals develop HCM or experience symptoms. Despite this highly variable phenotypic expression, it is generally accepted that variants in the gene for MYH7 are commonly correlated with higher penetrance of disease, earlier age at diagnosis, and more pronounced hypertrophy compared to mutations in MYBPC3. Despite this, studies comparing the two have offered contrasting results. In a cardiac magnetic resonance (CMR) study on 273 patients, impaired ventricular function was found to be more common among patients with MYBPC3 mutations, who showed higher susceptibility to arrhythmic events [81]. On the contrary, in a study that included 1316 patients, the severity of the HCM disease phenotype and clinical outcomes were consistent across individuals carrying pathogenic variants in MYBPC3 regardless of the type of variant (truncating or non-truncating), the location of the variant, or whether it was a founder or non-founder mutation. This highlights the importance of identifying additional genetic and non-genetic modifiers that can account for the diverse range of HCM disease phenotypes observed [54].

This variability is not limited to the phenotypic expression observed in patients with different causal mutations, and extends to family members who carry the same mutation. There are reported cases of identical twins sharing the same MYBPC3 mutation who have different phenotypes and disease severity. This further implies the possible impact of different unknown factors in the development of HCM [82]. In addition to HCM, identical MYBPC3 mutations can lead to left ventricular noncompaction and to restrictive or dilated cardiomyopathy [83]. MYBPC3 mutations were the second most commonly reported disease-causing mutations in a multi-center and multi-national study conducted in Europe that enrolled a cohort of 639 patients with sporadic or familial dilated cardiomyopathy [78]. Additionally, a frameshift-causing 25-base-pair deletion from intron 32 of MYBPC3, which was originally linked to HCM in two families, has been identified in 13.8% of patients with various inherited cardiomyopathies (including HCM and dilated and restrictive cardiomyopathy) [84,85].

There is a considerable amount of evidence indicating that patients who possess more than one mutation tend to develop a more severe phenotype. As opposed to heterozygous pathogenic mutations, homozygous or compound heterozygous truncating pathogenic MYBPC3 mutations can cause left ventricular noncompaction and septal defects, leading to severe neonatal cardiomyopathy [86].

There appear to be potential biomarkers for severe phenotypes of MYBPC3-associated HCM. Multiple pathways, including acylcarnitine, histidine, lysine, purine, steroid hormone metabolism, and proteolysis are altered in these patients, which could lead to possible risk stratification methods in the future [87].

### 4.3. Clinical Implications of Mutation Type

The type of mutation appears to be another important factor. Although the most common mutations are those causing loss of function, there are numerous patients with HCM and a MYBPC3 missense variant of uncertain significance (VUS). Individuals with MYBPC3 missense VUS predicted to disrupt subdomain folding (STRUM+) present a high incidence of adverse clinical outcomes, including ventricular arrhythmias, heart failure, all-cause mortality, atrial fibrillation, and stroke, similar to pathogenic MYBPC3 variants [88]. Various founder mutations with different phenotypic expressions have been identified. One example is the MYBPC3 c.2149-1G>A mutation, which has been shown to be more frequently associated with hypertrophy of the anterior interventricular septum, preserved left ventricular ejection fraction, and atrial enlargement. Male individuals who carry this mutation were found to exhibit higher penetrance and earlier disease onset [72]. At present, data supporting early recognition of cardiac impairment in asymptomatic carriers of MYBPC3 gene mutations are limited. A study using CMR imaging in asymptomatic females carrying a pathogenic gene variant did not identify any major reductions in myocardial contractile function [89]. However, individuals carrying the MYBPC3-Q10961X Finnish founder mutation exhibited heightened septal convexity regardless of whether or not they had LVH [90].

A characteristic of HCM is the impairment of myocardial oxygenation. In individuals with MYBPC3 HCM, this feature appears to be present during stress irrespective of diastolic function, left ventricular (LV) global longitudinal strain, or wall thickness [91].

Recent evidence suggests that asymptomatic carriers of MYBPC3^125bp^ mutation have a higher risk of left ventricular hypercontraction and diastolic dysfunction during the bicycle stress echocardiogram [92]. Studies focusing on genetic mutations and clinical correlations in HCM are summarized in Table 1.

### 4.4. Major Outcomes

The most extensive and highly reliable clinical datasets presently indicate that individuals diagnosed with HCM and possessing a pathogenic sarcomere mutation demonstrate heightened incidence of unfavorable clinical outcomes in comparison to HCM patients without pathogenic sarcomere gene mutations, regardless of the specific mutation type.

In a retrospective study on 211 Japanese patients with HCM, individuals with sarcomere gene mutations experienced disease-related morbid events at a significantly higher rate. Of note, the strongest association was with lethal arrhythmic events. Moreover, at the time of enrollment they had a higher occurrence of non-sustained ventricular tachycardia, displayed greater thickness of the interventricular wall, were more frequently in the dilated phase of HCM, and less frequently exhibited an apical pattern of hypertrophy [93].

Data characterized by meticulously curated genetic variants obtained from the Sarcomeric Human Cardiomyopathy Registry (SHaRe), a large-scale multicenter cohort, unequivocally support the conclusion that HCM attributed to sarcomere mutations is associated with more adverse outcomes compared to HCM not caused by such mutations. Patients carrying sarcomere mutations with compelling evidence of pathogenicity exhibited clinical manifestations of the disease at a younger age, experienced a greater burden of hypertrophic cardiomyopathy-related complications, and had earlier onset of these complications. The same analysis further revealed that the risk of developing the overall composite outcome by the age of 50 was 29% for patients with sarcomere mutations (SARC+), compared to 14% for patients without such mutations (SARC-). Even after adjusting for the earlier age at presentation, sarcomere mutations continued to confer a significant excess hazard. It was observed that patients with pathogenic or likely pathogenic sarcomere mutations faced a twofold greater risk of experiencing adverse outcomes and a fourfold higher risk of ventricular arrhythmias compared to those without any mutations. Furthermore, sarcomere variants classified as having uncertain significance were associated with an intermediate level of risk [95].

Moreover, there were differences in outcomes when comparing MYBPC3 to MYH7 mutations. In a study of 63 individuals with HCM (76% with MYBPC3 gene mutation), the patients in the MYBPC3 group more frequently had a positive family history of HCM, and dyspnea was the most prevalent symptom. On the contrary, the incidence of atrial fibrillation, systolic anterior motion of the mitral valve, and mitral leaflet abnormalities was more common in the MYH7 group [94]. Protein-truncating variants (PTVs) in MYBPC3 appear to favor worse outcomes, in particular those affecting the cardiac-specific M-motif at the N-terminus of the cMyBP-C protein, compared to the mutations in the non-cardiac regions [96].

## 5. Conclusions

Although knowing the pathogenic gene in HCM patients does not change the therapeutic course at the current time, genetic testing can offer prognostic data for both probands and their relatives. The specific type of mutation can predict the possible onset of the disease and the associated clinical outcomes.

However, current knowledge is insufficient to fully explain the highly variable phenotype of patients with MYBPC3 mutations, suggesting a possible interplay between the genetic variants and other modifiers that have not yet been fully described.

## Figures and Tables

**Figure 1 ijms-24-10510-f001:**
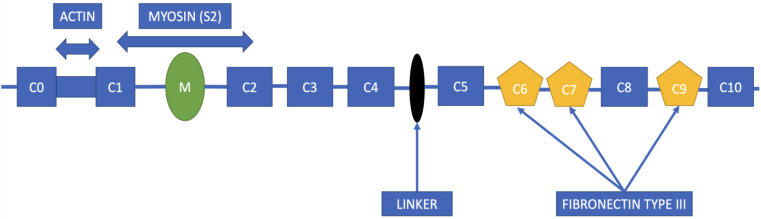
Schematic illustration of cMyBP-C protein structure, composed of eight immunoglobulin-like domains (C0, C1, C2, C3, C4, C5, C8, C10) and three fibronectin type III domains (C6, C7, C9). The interconnection between the C4 and C5 domains is denoted by a black oval.

**Figure 2 ijms-24-10510-f002:**
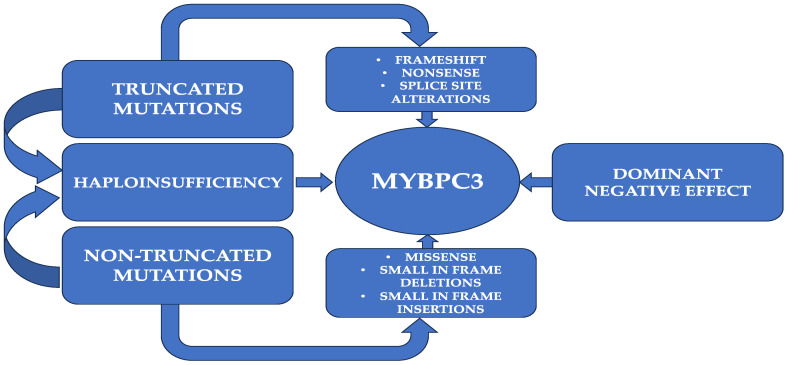
Schematic illustration of potential mutations that can occur at the MYBPC3 level.

**Table 1 ijms-24-10510-t001:** Studies of genetic mutations and clinical correlations in HCM.

Study	Country	Population	Results
Viswanathan et al. [6]	USA	150 individuals (80 symptomatic HCM patients, 35 asymptomatic carriers and 35 non-carriers)	HCM phenotype was independent of the specific causative mutation, despite MYH7 and MYBPC3 being the predominant ones
Page et al. [55]	UK	585 consecutive, unrelated individuals fulfilling diagnostic criteria for HCM	Families with MYBPC3 mutations exhibited diverse clinical presentations and prognosis, with no clear correlation between mutation type or specific mutation and the observed phenotype, indicating marked phenotypic variability within and between families sharing the same mutation.
Robyns et al. [73]	Belgium	378 HCM patients	Individuals carrying MYBPC3 mutations exhibited a higher incidence of sudden cardiac death in comparison to carriers of troponin complex mutations, MYH7 mutations and those without any mutations.
Lopes et al. [74]	Portugal	528 patients from the Portuguese Registry of Hypertrophic Cardiomyopathy (PRo-HCM)	Sarcomere-positive patients had distinct demographics, ECG, imaging features, family history and an increased risk of sudden cardiac death. The presence of a MYH7 mutation was associated with a progression towards left ventricular systolic dysfunction.
Terauchi et al. [75]	Japan	61 subjects (28 families) carrying MYBPC3 mutations	Females exhibited delayed onset of the disease, but increased symptomatology at the time of diagnosis and a higher frequency of heart failure events after the development of hypertrophy.
Adalsteinsdottir et al. [76]	Iceland	60 probands with HCM caused by MYBPC3 c.927-2A>G founder mutation and 225 first-degree relatives	Phenotypic expression was varied based on age, sex, and proband status. Men were more likely to develop left ventricular hypertrophy at a younger age and probands showed more prominent disease manifestations compared to relatives, with carriers without hypertrophy displaying subtle clinical differences from unaffected relatives.
Field et al. [77]	UK	62 consecutive patients diagnosed with HCM under 18 years of age and carrying at least one P/LP MYBPC3 variant	Significant and progressive LVH was observed in the cohort, with non-obstructive, arrhythmic disease phenotypes. Gender differences were observed, with worse clinical outcomes in male patients and earlier onset of the disease in males compared to females.
Miller et al. [81]	USA	273 HCM patients	Patients with MYBPC3 variants had a higher risk of impaired ventricular function and arrhythmic events, with a higher proportion receiving ICDs. Patients with identifiable gene variants had a higher LGE burden.
Helms et al. [54]	International	1316 HCM patients from the Sarcomeric Human Cardiomyopathy Registry	Cardiac structure and clinical outcomes were similar in patients with truncating versus nontruncating variants.
Dhandapany et al. [84]	India	800 cardiomyopathy cases and 699 controls	The 25-bp deletion was linked to a persistent risk of heart failure, delayed symptom onset, and mild hypertrophy.
Thompson et al. [88]	International	120 HCM patients from the Sarcomeric Human Cardiomyopathy Registry	Individuals carrying MYBPC3 missense STRUM+ VUS had a high incidence of adverse clinical outcomes, similar to those observed in patients with pathogenic MYBPC3 variants.
Méndez et al. [72]	Spain	79 HCM patients with a pathogenic MYBPC3 variant	Male carriers of the MYBPC3 c.2149-1G>A founder pathogenic variant had earlier onset and higher penetrance of HCM. Compared to MYBPC3 p.Arg502Trp/Gln carriers, they showed better LVEF and functional class and similar rates of major adverse outcomes.
Tarkiainen et al. [90]	Finland	67 individuals (47 subjects with the MYBPC3-Q1061X Finnish founder mutation and 20 healthy relatives without the mutation)	Subjects carrying the MYBPC3-Q10961X mutation showed higher septal convexity regardless of the presence of LVH.
Nakashima et al. [93]	Japan	211 HCM patients	Sarcomere gene mutation carriers experienced more morbid events, notably lethal arrhythmias. At enrollment, they showed a higher prevalence of non-sustained ventricular tachycardia, dilated HCM, increased interventricular wall thickness, and a lower incidence of apical hypertrophy.
Velicki et al. [94]	International	63 HCM patients (48 with a MYBPC3 mutation and 15 with a MYH7 mutation)	Patients with MYH7 gene mutations had greater disease severity compared to those with MYBPC3 mutations.
Ho et al. [95]	International	4591 HCM patients from the Sarcomeric Human Cardiomyopathy Registry	HCM patients with sarcomere mutations had more adverse outcomes compared to HCM patients without such mutations.

HCM, hypertrophic cardiomyopathy; ECG, electrocardiogram; LVH, left ventricular hypertrophy; ICD, implantable cardioverter-defibrillator; LGE, late gadolinium enhancement; VUS, variant of uncertain significance; LVEF, left ventricular ejection fraction.

## Data Availability

Not applicable.

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
