# Peer review of "An Update on MYBPC3 Gene Mutation in Hypertrophic Cardiomyopathy"

_ijms, 2023, doi:10.3390/ijms241310510_

Round 1

Reviewer 1 Report

The current review on MYBPC3 gene mutation in hypertrophic cardiomyopathy is well researched and well written. The authors did a commendable job. The review will be a great source of information for those interested in the specific field and to the wider public interested in cardiomyopathy.

Reviewer 2 Report

The authors Tudurachi et al have presented a review of MYBPC3 gene mutation in hypertrophic cardiomyopathy. This is an important topic of central relevance to the field. The authors do a nice job of summarizing available data and highlighting key areas for future investigations. However, there are several areas that require editing to improve clarity and more completely represent the most robust data available. This includes the discussion of central questions regarding the relationship of phenotype and genotype, gene structure and function, an MYBPC3 pathogenesis.  

Major comments:

1.       MYBPC3 gene structure and function

The authors nicely review the gene structure of MYBPC3 and regions of cMyBP-C that differ from skeletal muscle MYBPC isoforms. They also highlight key interactions within the sarcomere. However, cMyBp-C is referred to as a key structural component of the sarcomere. This is not the case, the sarcomere remains intact in the absence of cMyBP-C (Helms JCI insight 2020). Thus, the text should be edited to reflect that instead of being a key structural component making up a large part of the sarcomere,  cMyBP-C is a flexible rod-like protein that sits anti-parallel within the A-band of the cardiac sarcomere and regulates crosslinking kinetics through interactions with actin and myosin via it’s N-terminus and Titin and light meromyosin via it’s C-terminus. Also, in Figure 1 there is a black oval I believe is meant to highlight a linker region between C4 and C5 but is not well defined within the figure.

2.       MYBPC3 pathogenesis

The authors do a nice job of describing the various two pathogenic disease models (haploinsufficiency vs dominant negative) as well as associated pathways that could influence pathogenesis (UPS, NMD, and splicing defects). However, I would like to see improved clarity and organization in the discussion.

I recommend cutting figure 2 it falsely makes each of the mechanisms equal in importance and independent of one another.

In the text, when discussing truncating variants consider beginning presenting key concepts first. Such as that autosomal dominant inheritance of MYBPC3 variants may be explained by loss of function and haploinsufficiency or dominant negative poison peptide effect. Clearly define these terms; haploinsufficiency occurs when a single copy of the gene is functional and the other carries a loss of function mutation and disease is caused by a reduced level of functional gene product. Whereas a dominant negative mutation occurs when a gain of function pathogenic variant adversely affects function of remaining wild-type protein or another multimeric protein. You do this but you define these terms later in your discussion and I think providing some framework for your discussion could be helpful.

Next, I would highlight that primary experimental mechanism by which one can differentiate between these two pathogenic mechanisms is by evaluating how disease models respond to gene replacement therapy. Overexpression of wild-type MyBP-C should prevent disease phenotypes in models carrying pathogenic MYBPC3 variants if the model of haploinsufficiency is correct but would not be expected to have an effect if dominant negative poison peptide was the primary pathogenic mechanism. Conversely, overexpression of MYBPC3 carrying a truncating pathogenic mutation should not lead to disease phenotypes when expressed in a wild-type background if the model of haploinsufficiency is correct but would lead to disease phenotypes if dominant negative poison peptide was the primary pathogenic mechanism.  

After these concepts are introduced, then present the data for haploinsufficiency and pathways that can contribute to haploinsufficiency. Followed by data that may support a poison peptide hypothesis. The authors ultimately make the case that haploinsufficiency is the primary cause of HCM in MYBPC3 pathogenic mutants (line 321), please improve the clarity of this section to support this assertion and explain the thought process behind it. 

Some references not included to consider adding include;

MYBPC3 gene replacement experiments

-          Jiayang Li et all JCI Insight 2020 (doi: 10.1172/jci.insight.130182)

-          Mearini et al nature communications 2014 (doi: 10.1038/ncomms6515)

-          Paul JM Wijnker et al Journal of Molecular and Cellular Cardiology 2016 (doi: 10.1016/j.yjmcc.2016.03.003)

-          AM da Rocha et al Journal of Molecular and cellular cardiology 2016 (doi: 10.1016/j.yjmcc.2016.09.004)

3.       Figure 3 is conceptually identical to Figure 2 in Carrier Targeting the population for gene therapy with MYBPC3. Journal of  Molecular and Cellular Cardiology 2021. As such it should be cut. I encourage the authors to consider a unique figure that summarizes data surrounding evidence for haploinsufficiency (well supported hypothesis) and poison peptide (alternative hypothesis).

4.       Genotype and phenotype relationship

The authors describe a relationship between MYBPC3 mutation type and clinical phenotypes. While this remains an active area of research, the largest and most robust clinical data sets currently show that patients with HCM and a pathogenic sarcomere mutation (of any mutation type) exhibit a higher rate of adverse clinical outcomes than patients with HCM lacking pathogenic sarcomere gene mutations (Ho CY Circulation 2018). It has not yet been shown that the type of MYBPC3 mutation is associated with specific clinical outcomes. The authors should make efforts the clarify this throughout the text. I have highlighted a few examples of this with suggested edits below;  

Abstract

“The aforementioned genetic modifications, which appear to be associated with unfavorable lifelong outcomes and are largely influenced by the type of mutation, exhibit a unique array of clinical manifestations that range from asymptomatic to arrhythmic syncope or sudden cardiac death. Although the current understanding of the MYBPC3 mutation does not comprehensively explain the varied phenotypic manifestations witnessed in patients, the mutation's characteristics can function as a prognostic determinant for the onset of the disease and related clinical consequences”

Consider edit to
“The aforementioned genetic modifications, which appear to be associated with unfavorable lifelong outcomes and are largely influenced by the type of mutation, exhibit a unique array of clinical manifestations that range from asymptomatic to arrhythmic syncope or sudden cardiac death. Although The current understanding of the MYBPC3 mutation does not comprehensively explain the varied phenotypic manifestations witnessed in patients with HCM. Patients with pathogenic MYBPC3 mutations can exhibit an array of clinical manifestations ranging from asymptomatic, to advanced heart failure and sudden cardiac death.  , the mutation's characteristics can function as aHowever, it has been observed that patients with HCM and a pathogenic sarcomere gene mutation have a higher rate of adverse clinical outcomes than patients with HCM lacking pathogenic sarcomere gene mutations.  prognostic determinant for the onset of the disease and related clinical consequences”

Minor comments:

1.       Clarity

The structure of the first paragraph makes it difficult to follow. Given the focus on MYBPC3. I recommend the authors first define the prevalence of HCM and clinical manifestations. And then introduce the genetics in paragraph 2 (cutting mention of autosomal dominant inheritance from the first paragraph and adding information such as autosomal dominant inheritance of MYH and MYBPC3 specifically in paragraph 2.

Minor editing for grammar and clarity are needed throughout. 

Reviewer 3 Report

This manuscript perpetuates a common misunderstanding in hypertrophic cardiomyopathy (HCM), which is that this disease is primarily a genetic disorder (line 15-17). In reality the majority of patients with HCM do not have a gene mutation that can explain the cardiac hypertrophy. I agree that mutations in MYBPC3 are commonly the cause of saracomere gene mutation positive HCM, but in total it is responsible for ~20% of all HCM. This distinction should be made more clearly early in the manuscript – it should not require more than a sentence or two but it is critical for the reader to understand that actually the majority of HCM is sarcomere gene negative.

·         Line 44: MYBPC and MYH7 mutations account for 70% of sarcomere gene mutation positive HCM (not all HCM).

It is not clear whether MYBPC3 truncation mutations produce a truncated protein (line 19) or whether the premature stop codon results in non-sense mediated decay of the transcript resulting in a hypomorphic allele. While this point is made later (line 21-23) the strength of the evidence supporting an actual truncated protein being present in the hearts of patients with HCM is critical for our understanding of this form of genetic cardiomyopathy and needs to be better addressed.

“the mutation's characteristics can function as a prognostic determinant for the onset of the disease and related clinical consequences” (lines 27-29): This is a critical issue not well defended in the manuscript.

The role of MYBPC3 in DCM is less well established.

Line 58: the title of the manuscript is “Update”. This line should also say that this manuscript is an update and that only the newer information should be emphasized. After establishing what is the previous understanding of MYBPC3 and HCM the authors should then make clear in the manuscript what is an Update.

·         Has the new data actually supported the presence of a truncated protein being expressed?

In terms of clinical correlations with genetic status – the literature includes many studies describing small cohorts of individuals who due to referral bias give distorted impressions of the correlation of genetic status with clinical phenotype. It would be best to include a table that lists for each study the number of participants reported and the types of clinical phenotypes. Large cohorts should be emphasized.

The article should explore MYPBC3 mutations and their clinical correlation in non-European patients.

The quality of English language in the paper is appropriate - there are points where word choice could be improved or where there is unnecessary repetition that lengthens the manuscript but not necessarily in a manner that limits the overall readability of the paper.

Round 2

Reviewer 3 Report

The revised manuscript is significantly improved and provides the reader with a broader update on MYBPC and HCM.